# Episodic Memory in Amnestic Mild Cognitive Impairment (aMCI) and Alzheimer’s Disease Dementia (ADD): Using the “Doors and People” Tool to Differentiate between Early aMCI—Late aMCI—Mild ADD Diagnostic Groups

**DOI:** 10.3390/diagnostics12071768

**Published:** 2022-07-21

**Authors:** Athanasios Chatzikostopoulos, Despina Moraitou, Magdalini Tsolaki, Elvira Masoura, Georgia Papantoniou, Maria Sofologi, Vasileios Papaliagkas, Georgios Kougioumtzis, Efthymios Papatzikis

**Affiliations:** 1Neurosciences and Neurodegenerative Diseases, Postgraduate Course, Medical School, Faculty of Health Sciences, Aristotle University, 54124 Thessaloniki, Greece; tsolakim1@gmail.com; 2Greek Association of Alzheimer’s Disease and Related Disorders (GAADRD), 54643 Thessaloniki, Greece; 3Laboratory of Psychology, Department of Experimental and Cognitive Psychology, School of Psychology, Faculty of Philosophy, Aristotle University, 54124 Thessaloniki, Greece; emasoura@psy.auth.gr; 4Laboratory of Neurodegenerative Diseases, Center for Interdisciplinary Research and Innovation (CIRI-AUTH), Balkan Center, Aristotle University, 10th km Thessaloniki-Thermi, 54124 Thessaloniki, Greece; 5Laboratory of Psychology, Department of Early Childhood Education, School of Education, University of Ioannina, 45110 Ioannina, Greece; gpapanto@uoi.gr (G.P.); m.sofologi@uoi.gr (M.S.); 6Institute of Humanities and Social Sciences, University Research Centre of Ioannina (URCI), 45110 Ioannina, Greece; 7Department of Biomedical Sciences, School of Health Sciences, International Hellenic University, 57400 Thessaloniki, Greece; vpapaliagkas@gmail.com; 8Department of Turkish and Modern Asian Studies, National and Kapodistrian University of Athens, 15772 Athens, Greece; gkougioum@ppp.uoa.gr; 9Department of Early Childhood Education and Care, Oslo Metropolitan University, 0176 Oslo, Norway; efthymio@oslomet.no; 10Bright Start Foundation for Maternal and Child Health, 1209 Geneva, Switzerland

**Keywords:** verbal episodic memory, visuospatial episodic memory, recall, recognition, neuropsychological tool, Alzheimer’s disease

## Abstract

Episodic memory is the type of memory that allows the recollection of personal experiences containing information on what has happened and, also, where and when it happened. Because of its sensitivity to neurodegenerative diseases and the aging of the brain, it is considered a hallmark of Alzheimer’s disease dementia (ADD). The objective of the present study was to examine episodic memory in amnestic mild cognitive impairment (aMCI) and ADD. Patients with the diagnosis of early aMCI, late aMCI, and mild ADD were evaluated using the Doors and People tool which consists of four subtests examining different aspects of episodic memory. The statistical analysis with receiver operating characteristic curves (ROC) showed the discriminant potential and the cutoffs of every subtest. Overall, the evaluation of episodic memory with the Doors and People tool can discriminate with great sensitivity between the different groups of people with AD and, especially, early aMCI, late aMCI, and mild ADD patients.

## 1. Introduction

### 1.1. Amnestic Mild Cognitive Impairment and Alzheimer’s Disease Dementia

Alzheimer’s disease (AD) is a progressive neurodegenerative disease characterized by impairment in cognitive and functional abilities. The patients often have memory deficits, including impairment in learning and recall of recently gained information, and deficits in orientation, language, and visuospatial abilities [1].

Alzheimer’s disease dementia (ADD) is defined as a major neurocognitive disorder in the *Diagnostic and Statistical Manual of Mental Disorders*, 5th Edition (DSM-V). The diagnostic criteria include gradual and deteriorating impairment in two or more cognitive functions or behavior (attention, executive functions, learning, memory, language, perception, and social cognition) projected on standardized neuropsychological tests with significant deficits in functionality and the ability of completion of daily activities. These deficits can be supported with mutations in the amyloid precursor protein (APP), presenilin (PSEN) 1 and PSEN 2 genes [1]. For a more precise diagnosis, ADD can be distinguished in mild, moderate, or severe stages [2].

According to DSM-V, mild cognitive impairment (MCI) can be defined as an objective impairment in one or more cognitive functions projected on standardized neuropsychological tests, while maintaining the ability to complete daily activities [1]. MCI patients can also be divided in amnestic MCI with primary memory impairments and non-amnestic MCI with impairments in other cognitive functions except memory. These subtypes are further classified as “single domain” and “multi-domain”, depending on the impaired cognitive functions [3].

The MCI patients can also be divided into early and late stages. Patients in both stages meet the conventional criteria for MCI but early MCI reflects those at an earlier point in the clinical spectrum. The Alzheimer’s Disease Neuroimaging Initiative (ADNI) confirmed this classification using neuropsychological examination with two measures of language (animal fluency and Boston Naming Test), two measures of executive functions (Trail Making Test A and B) and the Rey Auditory Verbal Learning Test (number of words recalled and delayed free recall after 30 min). To confirm the results of the neuropsychological assessment, CSF biomarkers and cortical thickness biomarkers were also used. However, this study also noticed the need for further research to confirm this classification and its utility in clinical practice [4].

### 1.2. Episodic Memory

Episodic memory refers to the recollection of unique experiences in terms of their content (what), temporal occurrence (when), and location (where) [5]. A fundamental feature of episodic memory that has potential diagnostic value is associative memory. It is the ability to link the component parts via spatial, temporal, or other kinds of relationships to recreate a completed episode. This aspect of episodic memory is affected by neurodegeneration from the early stages of AD with a reduction in hippocampal and entorhinal cortex volumes [6,7]. The second important feature of episodic memory is spatial navigation. It is a process of determining and maintaining all the spatial information and orientation and it can be divided as allocentric (environment based) or egocentric (self-based) [6,8]. The ADD patients have major deficits in this field because they have difficulties in orientation that affect their daily activities [8].

Episodic memory is mediated by the circuity of the medial temporal lobe, including the hippocampus, which interacts with other cortical and subcortical structures [9]. Regarding its stages, the cortical structures are involved in many aspects of perception, whereas the medial temporal lobe has a key role in the organization and encoding of memories whose details are stored in those cortical areas. The retrieval stage is mediated by the hippocampus which combines information by the cortical areas to compose complete memories [10,11].

aMCI patients and ADD patients have difficulties in the retrieval stage due to the hippocampal dysfunction [12]. Specifically, MCI patients present increased right hippocampal activation during encoding and decreased left hippocampal and fusiform gyrus activation during retrieval [13]. Due to the progress of the disease, there are changes in connectivity of the hippocampus with the other regions of the brain, such as the sensory-motor, attention, visual, frontal, and subcortical regions [9]. To confirm these findings, Economou et al. (2016) used the Five-Words Test in ADD patients and noticed that they had encoding and retrieval difficulties even with five items, and their memory deficits were more significant in delayed free recall [14].

Depending on the content, episodic memory can be divided into verbal and visuospatial episodic memory, the former of which accesses verbal information and depends on language skills, and the latter includes information on position and the visual content of an event. The poor performance of ADD patients in verbal episodic memory tasks, such as the Rey auditory verbal learning test (RAVLT) and the California verbal learning test, was correlated with decreased volume of the left hippocampus and the decreased activation of medial temporal lobe [15,16]. Regarding the visuospatial episodic memory, Juncos-Rabadan et al. (2014) analyzed the specificity and sensitivity of the pattern recognition test (PRT), delayed matching to sample (DMS), and paired associated learning (PAL), which are visuospatial episodic memory tasks from the Cambridge Neuropsychological Test Automated Battery (CANTAB) and found they can sensitively discriminate aMCI patients from the healthy controls [17]. Moreover, Robinson et al. (2018) noticed that the assessment of visuospatial episodic memory with the memory circle test can predict the progression from MCI to ADD [18].

### 1.3. The Role of Episodic Memory in the Diagnosis of ADD

Decline in episodic memory is one of the hallmark symptoms of AD and a characteristic of aMCI [5]. For this reason, the measurement of episodic memory could be an appropriate predictor in the staging of AD. Episodic memory tasks, such as RAVLT and Rey–Osterrieth complex figure test (ROCFT), can predict the progression from aMCI to dementia with high levels of sensitivity [19]. The accuracy of these findings are confirmed with single-photon emission computed tomography (SPECT) data showing alterations in brain connectivity and cerebral blood flow, especially in the medial temporal lobe, and hypoperfusion in the posterior cingulate cortex [19,20]. Mascoso et al. (2019) examined 387 MCI amyloid-positive patients, cognitively staged as early or late based on episodic memory performance. With cross-sectional and longitudinal comparisons with amyloid, tau, and neurodegeneration markers and using the data and participants from ADNI’s database, it was proved that episodic memory can be useful in staging MCI and symptoms of prodromal ADD and completes the amyloid, tau, and neurodegeneration profile [21].

Furthermore, episodic memory could be a reliable predictor of daily functionality. Nakhla et al. (2021) evaluated episodic and semantic memory with an extensive neuropsychological examination and found that the episodic memory measure was the single best predictor of shopping ability in ADD patients [12]. Moreover, episodic memory is significantly impaired even from the early stages of the disease and these deficits are greater in AD than in MCI [22,23].

### 1.4. Measures of Episodic Memory—The Doors and People Tool

During an episodic retrieval task, people associate episodic memory, such as a word from a list, with particular details such as emotion, time, place, and other people [24]. Episodic memory is typically evaluated with tasks in which a set of words is presented to participants and they are requested to learn them. After a short break or an intervening task to allow the material to be removed from working memory, the participants complete a recall or a recognition test on the presented content. However, the most episodic memory tasks are focused on content, and they do not cover all the aspects of episodic memory. There is also a need for using tasks with ecological validity [6,25].

The use of naturalistic tasks in the clinical practice makes the patients feel more comfortable and imprints more precisely the deficits that affect their daily life [26]. For this reason, neuropsychological tests focused on ecological validity contain cognitive tasks that theoretically resemble cognitive activities of daily life [27]. According to this approach, the tasks should have high phenomenological validity and focus on the ability of the participant to cope with real world tasks regardless of the cause of the problem. Baddeley et al. (1994) created the Doors and People tool to cover the need for more ecologically valid tasks for episodic memory [28]. As a neuropsychological tool, it was developed to measure the real deficits of the patients but consists of more naturalistic material in comparison with other tasks. It includes stimuli familiar to the participants, to be accepted by a wide range of population, such as dementia and MCI patients and healthy people of any age, and not to cause anxiety in them.

There are many neuropsychological tasks measuring memory deficits but only a few of them facilitate the evaluation of the whole memory system analyzing the kind of memory deficits. The Doors and People tool was created to measure the kind of episodic memory deficits because it can make comparable measurements of visual and verbal episodic memory. Moreover, it includes recognition and free recall tasks [29]. Therefore, it was created as a response to the limitations of the already-in-use tasks of verbal and visuospatial episodic memory. Visuospatial measures, such as the ROCFT, include only one measurement in a complex figure evaluating not only episodic memory, but also perceptual ability and the ability of copying [30]. Verbal memory tasks may show ceiling effects when they are used in healthy people or in people with minor disorders and they have deficits in their phenomenological validity. In this way, low scores may be the result of perceptual deficits [30].

The Doors and People tool overcomes these restrictions. Specifically, in the subtests of verbal recall and recognition, full names were used, making the tasks ecologically valid and easily scored. Moreover, the choice of the stimuli was performed in such a way that the coding based on meaning or visual mental representations from names was reduced as far as possible. In the visual recognition test, images of doors were chosen because they are stimuli that the participants meet in their daily life and the use of distractors excludes the help from verbal signs. In the visual recall subtest, four simple variants of a cross are used, because no special skills are needed to draw them [29].

Another advantage of the Doors and People tool is that it can be used in a wide range of patient groups and in healthy people. It is a sensitive and useful tool that can be used from patients with mild memory deficits to patients with severe memory deficits due to dementia [31]. MacPherson et al. (2016) used it on patients with frontal lobe lesions to find the effects of these lesions on episodic memory performance in recall and recognition tasks [32]. Prokasheva et al. (2011) used it to measure episodic memory deficits following chemotherapy in breast cancer survivors [33], and Barbeu et al. (2011) administered it to examine the relationship of damage in the hippocampus with visual recognition [34]. Nestor et al. (2003) used it for differential diagnosis as they examined the differences between ADD and MCI patients. In that study, both groups displayed deficits in recall tasks whereas the ADD patients had worse performance [35]. Greene, Baddeley, and Hodges (1996), using the same tool, compared 33 early ADD patients with 30 matched controls to examine the causes of episodic memory deficits. The results showed that the deficits were caused by learning problems and not by oblivion or retrieval deficits [36]. It has also been used in studies with patients with epilepsy [37], schizophrenia [38], autism [39], and Asperger syndrome [40].

### 1.5. Aim and Hypotheses of the Study

Taking into consideration the lack of ecologically valid episodic memory tasks that reliably evaluate and cover more than one aspect of episodic memory, the primary objective of the present study was the discrimination between different groups (early aMCI, late aMCI, mild ADD) of people with AD, in terms of the rate of their cognitive impairment caused by AD, with the use of a specialized psychometric tool which measures episodic memory in many aspects, that is, the “Doors and People” tool.

The first hypothesis (H1) of the present study was that the evaluation of episodic memory using the Doors and People tool would reliably differentiate between early aMCI, late aMCI, and mild ADD people.

The second hypothesis (H2) was that the “recall” conditions of episodic memory as measured by the specific tool would more reliably differentiate between the three diagnostic groups, compared to “recognition” conditions.

The third hypothesis (H3) was that both the evaluation of visuospatial and verbal episodic memory would reliably differentiate between the three diagnostic groups.

## 2. Methods

### 2.1. Participants

The sample consisted of 90 Greek native speakers, 43 men and 47 women with a mean age of 77.06 (SD = 5.306) years and a mean education of 9.48 (SD = 3.302) years (Table 1). The assessment took place in the Greek Association of Alzheimer’s Disease and Related Disorders. The participants were divided into amnestic MCI (aMCI) patients (multidomain), who met the diagnostic criteria of Petersen et al. [2], and patients with mild AD (n = 30), who met the criteria of DSM-V [1]. The aMCI patients were, also, divided into early (n = 30) and late aMCI patients (n = 30) based on the deviations from cutoffs in the results from extended neuropsychological assessment, which includes tasks that examine many memory systems. The neuropsychological assessment included the following tests: Mini Mental State Examination (MMSE, cut-off score: 23/24) [41], Montreal Cognitive Assessment (MoCA, cut-of score: 26) [42], Trail Making Test A and B (TMT) [43], Rey Auditory Verbal Learning Test (RAVLT) [44], Rivermead Behavioral Memory Test (RBMT) [45], Functional Cognitive Assessment Scale (FUCAS cut-off: 45) [46], Rey Osterrieth Complex Figure (ROCFT) [47], phonemic verbal fluency [48] and Alzheimer’s Disease Assessment Scale-Cognitive Subscale (ADAS-COG) [49]. Moreover, depression was measured with the Geriatric Depression Scale (GDS, cut-off score: 6–7) [50] and anxiety was measured using the Short Anxiety Screening Test (SAST, cut-off score: 22–23) [51].

Univariate analysis of variance (ANOVA) was conducted for the examination of potential differences of the three diagnostic groups in age and education. There were statistically significant effects of the diagnostic group on age, F(2, 87) = 43.242, *p* < 0.001. As far as education is concerned, there were no statistically significant effects of the diagnostic group on years of education, F(2, 87) = 3.018, *p* = 0.054. Chi-square analysis, as regards gender and diagnostic group, showed that there were no statistically significant differences between the groups, χ^2^(2, 90) = 0.043, *p* = 0.958.

All three groups of the sample followed the same processes, which included clinical examination, laboratory/imaging procedures and neuropsychological assessment. The results were evaluated by the neurologist of the Greek Association of Alzheimer’s Disease and Related Disorders (GAADRD). The exclusion criteria of the study were: (1) hearing deficits that could affect the performance of episodic memory, (2) uncorrected visual impairment, (3) inability to comprehend Greek language, (4) currently taking antidepressant medication, anxiolytics, or mood stabilizers, and (5) currently taking antipsychotic medication.

### 2.2. Procedure

All the participants of the study had been examined by a neurologist with neurological examination and a neuropsychologist with neuropsychological assessment. After the diagnosis, the participants were asked to participate voluntarily in the study. The administration of the Doors and People tool took place in the Greek Association of Alzheimer’s Disease and Related Disorders. It was conducted individually for each patient in a soundproof room by a trained psychologist and the duration of the assessment was 40–50 min.

### 2.3. Ethics

All study participants read the information sheet and signed the informed consent during the initial clinical visit, stating that the research group of the GAADRD have the permission to use their demographic data, which would be anonymized, such as gender, age, and education, as well as their performance in the neuropsychological tests, for research purposes. For participants with dementia in mild stages, a legal representative read the consent and signed the relevant document.

The study was approved by the Scientific and Ethics Committee of the GAADRD (Scientific Committee Approved Meeting Number: 67-4/17-04-2021), which follows the new General Data Protection Regulation (EU) 2016/679 of the European Parliament and of the Council of 27 April 2016 on the protection of natural persons with regards to the processing of personal data and on the free movement of such data, as well as the principles outlined in the Helsinki Declaration.

### 2.4. Measure

*Doors and People: A Test of Visual and Verbal Recall and Recognition*: It is a test of episodic memory and consists of four subtests [27,28]. Each subtest has ecological validity, and it covers a wide range of participant groups including MCI and dementia [28,29]. Furthermore, the test has been adapted and validated in the Greek population. In the Greek version there was statistically significant correlation between the subtests, a negative correlation with age meaning that age affects the scores of the subtests, and a positive correlation with educational level. The test including its four subtests has satisfactory internal validity (Cronbach’s α = 0.803) showing that it is a reliable tool, and the content was carefully chosen to have ecological validity [30].

The People test measures immediate and delayed verbal recall. The stimuli consist of the photographs of four characters with their names and occupation printed below. They are presented for 3 seconds, and the character name and occupation are read aloud (e.g., This is a doctor. His name is Hλίας Τσακίρης (Elias Tsakiris). The presentation is repeated until all four names can be correctly recalled (a maximum of three trials). The participant is requested to recall this information immediately after the presentation and after 5–10 min. One point is awarded for each correct first name and surname and an additional point for each correct pairing. The total score is obtained from the sum of every trial’s individual score (score range: 0–36) [29]. In the Greek validation of the People subtest, the mean score of the healthy adults with age between 65–80 years old was 15.2 (SD = 4.3) [31].

The Doors test measures visual recognition. In this test, 24 pictures of doors separated in two sets (an easy set and a hard set) are presented to the participant. After the presentation, the participant is called to choose among four doors (3 distractors and the target door) the door that was presented before. In the first set (Part A) the distractors are different categories of doors in comparison with the target door (e.g., a garage door, a French door, a front door) whereas in the second set (Part B) the distractors are the same type of door (e.g., all stable doors). One point is awarded for each correct response and the total score consists of the sum of the scores of each set (score range: 0–24) [29,30]. In the Greek validation of the Doors subtest, the mean score of the healthy adults with age between 65–80 years old was 12.5 (SD = 3.3) [31].

The Shapes test measures immediate and delayed visual recall. Four-line drawings of crosses are presented, and the participant is called to draw them immediately after the presentation and after 5–10 min. The shapes are presented until the participant will be able to recall them correctly (a maximum of three trials). Each correct shape is awarded with three points and the total score is obtained from the sum of every trial’s individual score (score range: 0–36) [29,30]. In the Greek validation of the Shapes subtest, the mean score of the healthy adults with age between 65–80 years old was 27 (SD = 4.5) [31].

The Names test measures verbal recognition. Twenty-four names (both forename and surname) separated in two sets (an easy set and a hard set) are presented for 3 seconds each and the participant is asked to read them out loud. After the presentation, the participant is called to choose among four names (3 distractors and the target name) the name that was presented before. Τhe second set (Part B) consists of names where the distractors are more similar to the target name. One point is awarded for each correct response and the total score derives from the sum of the scores of each set (score range: 0–24) [29]. In the Greek validation of the Names subtest, the mean score of the healthy adults with age between 65–80 years old was 13.9 (SD = 3) [31].

### 2.5. Statistics

The statistical analysis was performed with the use of IBM SPSS Statistics, version 27 and the statistical significance was set at 0.05. Multivariate analysis of variance was conducted for the examination of the effect of age depending on the diagnosis on the Doors and People tool and subtests. Receiver operating characteristic (ROC) curve analysis was conducted to assess the predictive value of every subtest of the Doors and People tool to discriminate early aMCI patients from late aMCI patients and mild ADD patients. Finally, sensitivity and specificity were also extracted to evaluate the ability of the Doors and People tool to discriminate the three groups by calculating the cutoffs of every subtest. The area under the curve (AUC) was used to quantify the discriminant potential of every subtest into poor (0.51–0.69), fair (0.70–0.79), good (0.80–0.89), excellent (0.90–0.99) and perfect (1.0) [52]. The cutoff points were determined by maximizing the Youden index.

## 3. Results

At first, mean scores of every subtest in the three diagnostic groups were calculated (Figure 1).

Multivariate analysis of variance (MANOVA) was conducted to identify the effects of age group on Doors and People scores based on diagnosis (Table 2). We created two age-category groups. The first age group consisted of participants with age-range 60–75 years old and the second age-group had participants with age-range 76–90 years old. Next, a two (age-group) × three (diagnostic group) between-subjects multivariate analysis of variance was performed on four dependent variables which were the scores of the four subtests (People, Doors, Shapes, Names). The analysis showed only a statistically significant main effect of the diagnostic group on the test, F(8, 162) = 3.907, *p* < 0.001 (Table 3). Moreover, Table 4 shows the effects of the diagnostic group on each one of the four subtests.

Predictive validity was assessed using ROC curves to estimate the score of every subtest of the Doors and People tool among the three groups of the sample. The ROC curves of the scores of the three groups are shown in Figure 2, Figure 3 and Figure 4, and the sensitivity, the specificity, and the cutoffs are shown in Table 4.

The ROC curves in Figure 2 show the score of every subtest between early aMCI and late aMCI patients. The discriminant potential of the People subtest is good (AUC = 0.803, SD = 0.060, *p* < 0.01), of the Doors subtest is fair (AUC = 0.763, SD = 0.062, *p* < 0.01), of the Shapes subtest is good (AUC = 0.854, SD = 0.051, *p* < 0.01) and of the Names Subtest is fair (AUC = 0.794, SD = 0.059, *p* < 0.01).

The ROC curves in Figure 3 show the score of every subtest between late aMCI and mild ADD patients. The discriminant potential of the People subtest is good (AUC = 0.858, SD = 0.048, *p* < 0.01), of the Doors subtest is poor (AUC = 0.636, SD = 0.071, *p* = 0.70), of the Shapes subtest is good (AUC = 0.825, SD= 0.052, *p* < 0.01), and for the Names subtest is poor (AUC = 0.603, SD = 0.074, *p* = 0.171).

The ROC curves in Figure 4 show the score of every subtest between early aMCI and mild ADD patients. The discriminant potential of the People subtest is excellent (AUC = 0.938, SD = 0.033, *p* < 0.01), of the Doors subtest is good (AUC = 0.854, SD = 0.051, *p* < 0.01), for the Shapes subtest is excellent (AUC = 0.962 SD= 0.023, *p* < 0.01) and for the Names subtest is good (AUC = 0.834, SD = 0.054, *p* < 0.01).

## 4. Discussion

Early diagnosis of MCI is considered as a major challenge in clinical practice. Knowing that episodic memory deficits are usual symptoms of dementia, the present study was focused on the role of episodic memory in the differentiation between early aMCI, late aMCI, and mild ADD patients. To do that, the Doors and People tool was used because it evaluates both visual and verbal aspects of episodic memory and has ecological validity.

Because of the extended range of aMCI, the classification in early and late aMCI makes the diagnosis more precise, leading to more appropriate clinical interventions. In the present study, the testing of episodic memory has been proven able to differentiate early aMCI and late aMCI groups. Specifically, the early aMCI patients had significant differences in all aspects of episodic memory in comparison with the late aMCI patients. The People and Shapes subtests had better discriminant potential than the Doors and Names, which had fair discriminant potential, showing that the “recall” tasks are more reliable ”differentiators” than the “recognition” tasks, confirming in this way the second hypothesis of the study. The third hypothesis was also confirmed because the results showed that both the evaluation of visuospatial and verbal episodic memory can discriminate early from late aMCI participants. These results could be explained by progressively extensive damage in the hippocampus. Lee et al. (2017) examined healthy, early MCI, and late MCI participants. The hippocampal shape modeling based on progressive template surface deformation on T1-weighted MRI images, showed significant atrophy in the bilateral Cornu Ammonis (CA)1 regions and the right ventral subiculum in early MCI compared with widespread atrophy of the hippocampus in late MCI participants. The analysis of diffusion tensor imaging (DTI) and Voxel-based morphometry (VBM) showed increased diffusivity in CA2-CA4 regions in early MCI and additionally in the subiculum region in late MCI participants. Moreover, the DTI results indicated that specific regions of the right hippocampus are affected earlier than those in the left hippocampus [53]. Kang et al. (2019) also compared these three groups with brain Voxel morphometry. The early MCI participants displayed reduced gray matter volume in the right middle temporal gyrus compared with healthy participants. The late MCI participants showed atrophy in the left parahippocampal gyrus, in comparison with the healthy participants, and reduced gray matter volume in the left fusiform gyrus in comparison with early MCI participants [54]. Moreover, the findings of the present study are in accordance with ADNI’s discrimination of MCI with the use of RAVLT which is a “recall” task assessing only verbal episodic memory [4].

However, the discrimination between late aMCI patients and mild ADD patients is more difficult. In this case, only verbal recall, measured by the People subtest, and visual recall, measured by the Shapes subtest, had good discriminant potential between these two groups, but the “recognition” tasks (Doors and Names subtest) had poor discriminant potential, clearly confirming the second hypothesis. According to the present results, both verbal and visuospatial episodic memory tasks are reliable ”differentiators” as long as “recall” conditions are used, confirming the first and third hypotheses, too.

Furthermore, the evaluation of episodic memory can differentiate between early aMCI and mild ADD patients. All aspects of episodic memory can reliably discriminate the early aMCI patients from mild ADD patients, confirming the first hypothesis. Specifically, the “recall” tasks (People and Shapes subtest) had excellent discriminant potential and the “recognition” tasks (Doors and Names subtest) had good discriminant potential, confirming the second hypothesis.

Over the last years, studies with different methodological design have been conducted to examine episodic memory deficits in older adults and the predictive role of episodic memory in the conversion of MCI to ADD. The study of Perry et al. (2007) showed that the tests, which assess the delayed free recall of verbal and visual information, have great sensitivity in predicting the conversion from MCI to ADD [55]. Moreover, Boraxbekk et al. (2015), using sentence learning with and without enactment in a longitudinal cohort, proved that free recall-based tests of episodic memory are useful for detecting individuals at risk of developing dementia up to 10 years prior to the diagnosis [56].

Overall, the present study confirmed preliminary results showing that MCI patients had significant deficits in episodic memory tests of acquisition, delayed recall, and associative memory. Specifically, a Face-Name pairs test, similar to the People subtest, may have the potential to be a useful neuropsychological tool for the identification of the MCI and the prediction of conversion to probable ADD within two years [57]. The study of Warren et al. (2021) also showed that episodic memory measures, such as the logical memory test (LMT) and RAVLT, can be reliable predictors of MCI to mild ADD progression [58]. Furthermore, the present results are supported by other studies which used longitudinal design [55,59,60]. De Simone et al. (2019) found that aMCI patients who converted to ADD during a 3 year follow-up displayed worse performance than healthy controls and non-converter MCI patients both in recall and recognition of the 15-word learning test [60]. Marra et al. (2016), using RAVLT and ROCFT, constructed an episodic memory score and evaluated aMCI converters to dementia and aMCI stable patients in a 2 year-follow-up. The ROC curves and the regression analysis showed that the episodic memory score is a reliable predictor of progression from aMCI to dementia [61].

Moreover, the significance of verbal episodic memory has been confirmed with MRI and Voxel-based morphometry. Specifically, the loss of hippocampal gray matter can be a predictor for conversion from MCI to ADD. [62,63]. Leube et al. (2008), comparing healthy subjects, MCI, and ADD patients, found that a gradual loss of hippocampal gray matter is correlated with deficits in verbal episodic memory measured using the verbal learning memory test (VLMT) [63]. Chang et al. (2010), using the RAVLT, examined the role of ability of learning and retention in the conversion from MCI to ADD over a 2 year-follow up. The results showed that the poor performance on the test is considered a prognostic marker for ADD and learning deficits are associated with a more widespread pattern of gray matter loss whereas retention deficits are associated with a more focal gray matter loss mainly in medial temporal regions [64]. For this reason, the evaluation of episodic memory can be a strong predictor of MCI to ADD progression in combination with measurements of cortical thickness, hippocampal atrophy and the beta-amyloid (Ab) [65,66]. Eckerstrom et al. (2013) proved that the RAVLT is the best individual predictor (AUC = 0.93). Its combination with hippocampal volume and the CSF biomarkers Aβ amyloid, P-Tau and T-Tau improved the predictive validity (AUC = 0.96) [67].

A limitation of the present study is that the participants in each group have significantly different ages. However, this could be explained by the fact that ADD is an age-related neurodegenerative disease [68,69,70], so the early aMCI participants were slightly younger than the late aMCI participants and the late aMCI were younger than mild ADD participants, respectively. For this reason, a suggestion for further research is a study with longitudinal design where the participants will be reevaluated with the Doors and People tool after some years from the baseline assessment. Moreover, the use of neuroimaging methods, such as MRI or SPECT, cortical thickness measurements and VBM, and the use of CSF biomarkers could strengthen the present data.

In conclusion, the present study, using a detailed separation of aMCI group, proved that the discrimination between AD patients with different rates of cognitive impairment, namely, early aMCI, late aMCI, and mild ADD patients, can be achieved with the evaluation of episodic memory using the Doors and People tool. This tool can be a reliable neuropsychological task. In contrast to tasks already in use, the Doors and People tool provides a complete evaluation of episodic memory using naturalistic tasks and evaluating all aspects of episodic memory with both “recall” and “recognition” conditions. All subtests can reliably discriminate early aMCI from late aMCI patients. The People and Shapes subtest, which contains “recall” tasks, can also discriminate late aMCI from mild ADD patients. The Doors and People tool is used in a wide range of neurocognitive disorders for clinical and research purposes. However, the present study proved that it can be used in aMCI and mild ADD patients, adding sensitivity, specificity, and validity to the clinical examination of memory.

## Figures and Tables

**Figure 1 diagnostics-12-01768-f001:**
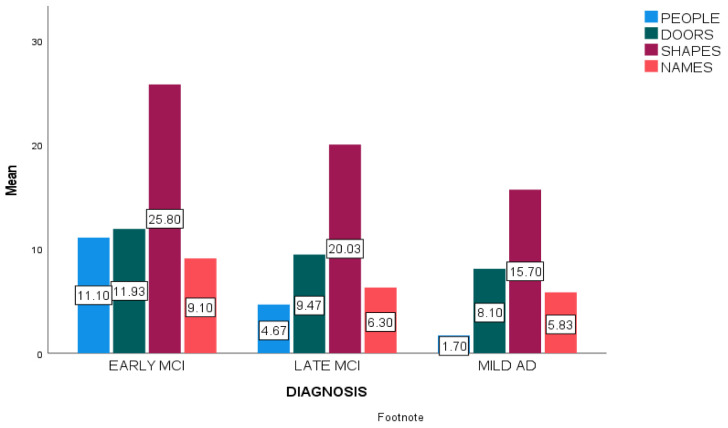
Mean scores of Doors and People subtests in early aMCI, late aMCI and mild ADD participants.

**Figure 2 diagnostics-12-01768-f002:**
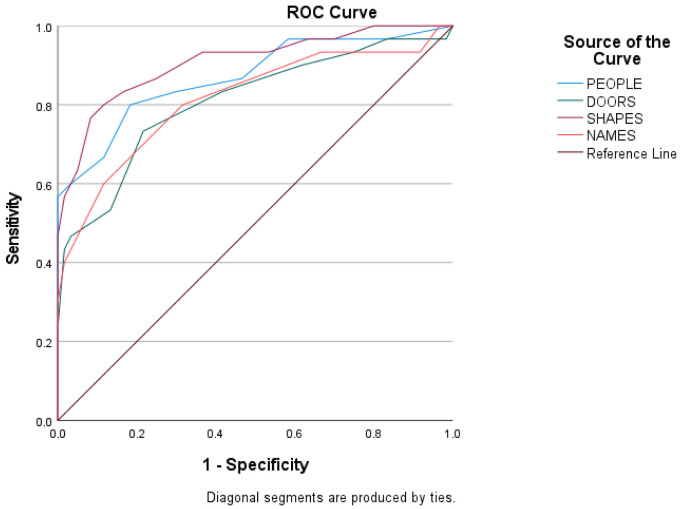
Subtests’ index score between early aMCI and late aMCI group.

**Figure 3 diagnostics-12-01768-f003:**
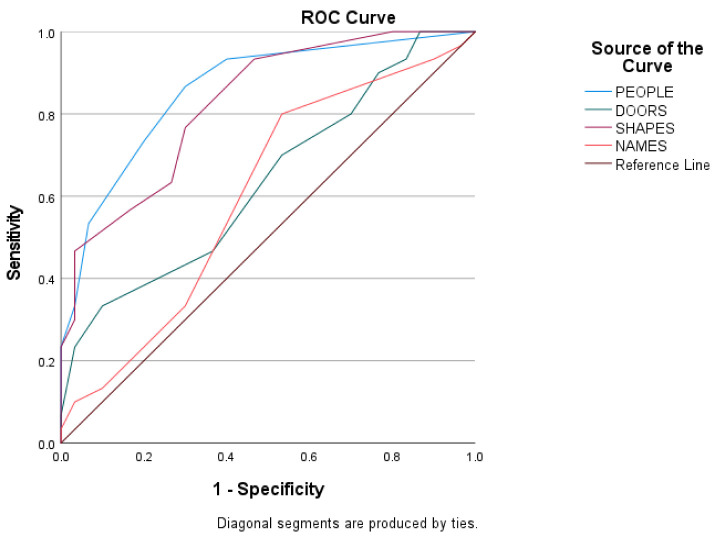
Subtests’ index score between late aMCI and mild ADD group.

**Figure 4 diagnostics-12-01768-f004:**
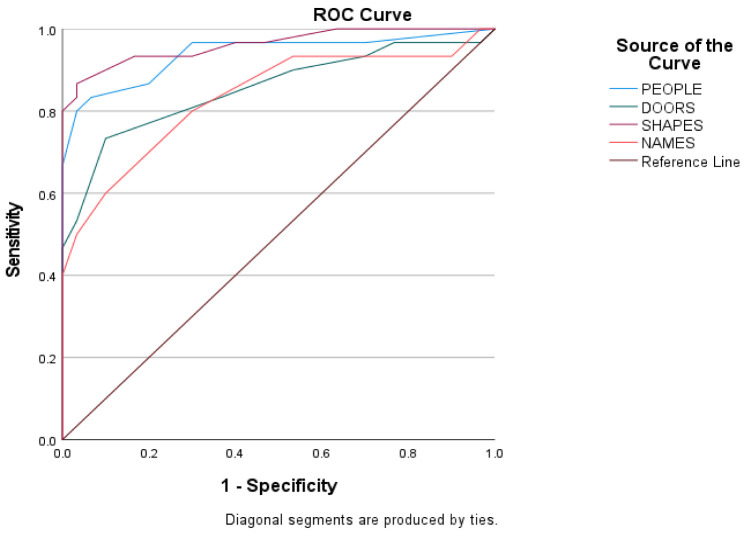
Subtests’ index score between early aMCI and mild ADD group.

**Table 1 diagnostics-12-01768-t001:** Demographics.

	Gender		Age		Education	
	% Male	% Female	M	SD	M	SD
Early aMCI	46.66	53.34	71.87	5.002	10.57	3.785
Late aMCI	50	50	78.93	3.073	8.53	2.763
Mild ADD	46.66	53.34	80.40	2.978	9.33	3.055

**Table 2 diagnostics-12-01768-t002:** Effects of age-group and diagnostic group on the Doors and People tool according to the MANOVA.

	Value	F	Df	*p*
Age-group	0.954	0.970	4	0.428
Diagnostic group	0.703	3.907	8	<0.001 **
Age-group × Diagnostic group	0.887	1.254	8	0.271

** *p* < 0.01.

**Table 3 diagnostics-12-01768-t003:** Effects of diagnostic group on the Doors and People subtests according to a series of ANOVAs.

	F	Df	*p*	Partial n^2^
People	12.616	2	<0.001 **	0.231
Doors	5.694	2	0.005 **	0.119
Shapes	9.323	2	<0.001 **	0.182
Names	4.834	2	0.010 *	0.103

* *p* < 0.05; ** *p* < 0.01. Post-hoc analysis using Scheffe criterion indicated statistically significant differences in the People subtest comparing early aMCI with late aMCI (I-J = 6.43, *p* < 0.001) and mild ADD participants (I-J = 9.40, *p* < 0.001). The comparison of late aMCI with mild ADD indicated, also, statistically significant differences (I-J = 2.97, *p* = 0.019). In the Doors subtest, there were statistically significant differences comparing early aMCI with late aMCI (I-J = 3.03, *p* < 0.001) and mild ADD participants (I-J = 4.40, *p* < 0.001). However, this subtest was not able to differentiate between late aMCI and mild ADD groups. In the Shapes subtest, post-hoc analysis indicated statistically significant differences comparing early aMCI with late aMCI (I-J = 5.77, *p* < 0.001) and mild ADD participants (I-J = 10.10, *p* < 0.001). Moreover, there were significant differences between late aMCI and mild ADD participants (I-J = 4.33, *p* < 0.001). In the Names subtest, there were statistically significant differences only in the comparison of early aMCI with late aMCI (I-J = 2.80, *p* < 0.001) and mild ADD participants (I-J = 3.27, *p* < 0.001).

**Table 4 diagnostics-12-01768-t004:** Diagnostic classification between the groups of Early aMCI, Late aMCI, and Mild ADD.

		Cut-Offs	Sensitivity %	Specificity %	*p*
Early MCI-Late MCI	People	5.5	80	66.7	<0.01 **
	Doors	10.5	73.3	77.7	<0.01 **
	Shapes	21.5	83.3	70	<0.01 **
	Names	6.5	80	66.7	<0.01 **
Late MCI-Mild AD	People	2.5	86.7	70	<0.01 **
	Doors	_	_	_	
	Shapes	17.5	76.7	70	<0.01 **
	Names	_	_	_	
Early MCI-Mild AD	People	4.5	83.3	93.3	<0.01 **
	Doors	9.5	83.3	63.3	<0.01 **
	Shapes	19.5	93.3	83.3	<0.01 **
	Names	6.5	80	70	<0.01 **

** *p* < 0.01.

## Data Availability

Data available upon duly justified request.

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
