# Peer review of "Episodic Memory in Amnestic Mild Cognitive Impairment (aMCI) and Alzheimer’s Disease Dementia (ADD): Using the “Doors and People” Tool to Differentiate between Early aMCI—Late aMCI—Mild ADD Diagnostic Groups"

_diagnostics, 2022, doi:10.3390/diagnostics12071768_

Round 1
Reviewer 1 Report
The objective of the present study was to find the predictive value of episodic memory in the progression of amnestic Mild Cognitive Impairment (aMCI) to Alzheimer’s disease dementia (ADD). Patients with the diagnosis of early aMCI, late aMCI and mild ADD were evaluated using the Doors and People tool which consists of four subtests examining different aspects of episodic memory. The statistical analysis with ROC curves showed the discriminant potential and the cutoffs of every subtest. Overall, the evaluation of episodic memory with Doors and People tool can discriminate with great sensitivity the different stages of ADD and predict the progression of aMCI to ADD.
This manuscript is well written and I have some concerns:
1. In Fig. 1: the mean score of the healthy participants should be included.
2. The calculation of discriminant potential should be provided.
3. Page 5, line 5-6: “mean age of 70.07” should be “mean age of 77.06”, as calculated from the data in Table I.
Minor Essential Revisions
1. In ABSTRACT section: “ROC curves” should be “Receiver operating characteristic curve (ROC)”.
2. Page 2, line 4: “DSM-V” should be “Diagnostic and Statistical Manual of Mental Disorders, 5th Edition (DSM-V).
3. Page 2, line 8: “APP, PSEN 1” should be “amyloid precursor protein (APP), presenilin (PSEN) 1”.
4. Page 2, line 10: “stage” should be “stages”.
5. Page 2, line 14, 18: “can, also, be divided” should be rewritten as “can also be divided”.
6. Page 3, line 21: “Rey Osterrieth Figure Test” should be “Rey Osterrieth complex Figure Test (ROCFT)”.
7. Page 3, line 22: “SPECT” should be “Single-photon emission computed tomography (SPECT)”.
8. Page 5, line 10: “DSM V” should be “DSM-V”.
9. Page 5, line 19-20: “The Alzheimer's Disease Assessment Scale (ADAS-COG)” should be “The Alzheimer's Disease Assessment Scale-Cognitive Subscale (ADAS-COG)”.
10. Page 5, lin40-41: “(GAADRD)” should be transferred into line 32.
11. Page 11, line 11: “CA1” should be “Cornu Ammonis (CA)1”.
12. Page 11, line 13: “DTI and VBM” should be “diffusion tensor imaging (DTI) and voxel based morphometry (VBM)”.
13. Page 11, line 30, 33: “1rst” should be “1st”.
14. Page 12, line 18: “beta-amyloid” should be “beta-amyloid (Ab)”.
Author Response
Dear Reviewer,
Thank you very much for your review and your detailed report. All the corrections made according to your suggestions are pointed with red colour in the text using the "Track Changes" function in MS Word and they are listed below with some comments from our site:
- "In Fig. 1: the mean score of the healthy participants should be included". Thank you for your revision. We cannot include the mean scores of healthy participants in the Figure 1 because in the present study there was not a group with healthy participants. Therefore, the present study is reffered to early aMCI, late aMCI and mild ADD patients. The mean scores of healthy adults with age between 65-80 have been published in another study by other researchers and they are mentioned in pages 6 and 7 where every subtest is presented (page 6 lines 37, 48 and page 7 lines 3,12).
- "The calculation of discriminant potential should be provided". Thank you for your revision. In the Statistics section in page 7, line 26, we mention that "The Area Under the Curve (AUC) was used to quantify the discriminant potential of every subtest into poor (0.51-0.69), fair (0.70-0.79), good (0.80-0.89), excellent (0.90-0.99) and perfect (1.0)". We, also, added that "The cutoff points were determined by maximizing the Youden index"
- "Page 5, line 5-6: “mean age of 70.07” should be “mean age of 77.06”, as calculated from the data in Table I". Thank you for your revision. We have corrected it.
- "Minor Essential Revisions" Thank you for pointing all these minor essential revisions. We have corrected them all and they are pointed with red colour.
Thank you very much for your revisions.
Best regards!
Reviewer 2 Report
Authors tested episodic memory of patients diagnosed of early aMCI, late aMCI and mild ADD using “Doors and People tool” test. They used MANOVA and ROC curves of the 4 subsets to compare the three diagnostic groups. They concluded that Doors and People tool can reliably discriminate different stages of ADD and predict the progression of aMCI to ADD.
The design and methods are appropriate, and results are very helpful for clinic. Authors discussed the details about their study.
Author Response
Dear Reviewer,
Thank you very much for your review and your commends.
Best regards